# PLA-PEG Implant as a Drug Delivery System in Glaucoma Surgery: Experimental Study

**DOI:** 10.3390/polym14163419

**Published:** 2022-08-21

**Authors:** Viktoriya N. Germanova, Elena V. Karlova, Larisa T. Volova, Andrey V. Zolotarev, Viktoriya V. Rossinskaya, Ivan D. Zakharov, Aleksandr R. Korigodskiy, Violetta V. Boltovskaya, Irina F. Nefedova, Mariya V. Radaykina

**Affiliations:** 1Department of Ophthalmology, Samara State Medical University, 443068 Samara, Russia; 2Eroshevskiy Eye Hospital, 443068 Samara, Russia; 3Biotechnology Center “BioTech”, Samara State Medical University, 443079 Samara, Russia; 4HiBiTech, 129110 Moscow, Russia; 5Institute of Experimental Medicine and Biotechnology, Samara State Medical University, 443079 Samara, Russia

**Keywords:** drug delivery system, glaucoma filtration surgery, wound healing modulation, cyclosporine A, everolimus, poly(lactic acid) implant

## Abstract

Excessive postoperative scarring halts the effectiveness of glaucoma surgery and still remains a challenging problem. The purpose of this study was to develop a PLA-PEG-based drug delivery system with cyclosporine A or everolimus for wound healing modulation. Methods: PLA-PEG implants saturation with cyclosporine A or everolimus as well as their further in vitro release were analyzed. Anti-proliferative activity and cytotoxicity of the immunosuppressants were studied in vitro using human Tenon’s fibroblasts. Thirty-six rabbits underwent glaucoma filtration surgery with the application of sham implants or samples saturated with cyclosporine A or everolimus. The follow-up period was six months. A morphological study of the surgery area was also performed at seven days, one, and six months post-op. Results: PLA-PEG implants revealed a satisfactory ability to cumulate either cyclosporine A or everolimus. The most continuous period of cyclosporine A and everolimus desorption was 7 and 13 days, respectively. Immunosuppressants demonstrated marked anti-proliferative effect regarding human Tenon’s fibroblasts without signs of cytotoxicity at concentrations provided by the implants. Application of PLA-PEG implants saturated with immunosuppressants improved in vivo glaucoma surgery outcomes. Conclusions: Prolonged delivery of either cyclosporine A or everolimus by means of PLA-PEG implants represents a promising strategy of wound healing modulation in glaucoma filtration surgery.

## 1. Introduction

Despite major advances in pharmacology, issues regarding drug delivery still remain quite challenging in different fields of medicine. In particular, glaucoma surgery still demands new developments in wound healing modulation which is usually performed by means of various implants, drainage devices, and therapeutic agents (or their combination) [1,2,3]. 

Glaucoma is a chronic progressive disease leading potentially to irreversible blindness. The number of people suffering from glaucoma is about 70 million worldwide. By 2040, this number is expected to surge to 111.8 million [4]. 

Glaucoma surgery is considered to be the most effective way to lower intraocular pressure (IOP)–a key modifiable risk factor of disease progression [5,6,7]. Outcomes of glaucoma filtration surgery, regardless of its type, depend mostly on the intensity of postoperative scarring [8,9]. Accomplished postoperative wound repair is undesirable in order to maintain surgically created aqueous humor outflow pathways.

Despite the fact that there have been proposed numerous methods of wound healing modulation considering application of glaucoma drainage devices, implants and various drugs, the problem of postoperative scarring in glaucoma surgery is still challenging. It is likely that one of the reasons for this is the major advance in hypotensive topical treatment which effects eye surface and in case of long-term pre-surgical medication promotes inflammation and excessive postoperative scarring [10,11]. 

Among pharmacological agents used for wound healing modulation in glaucoma surgery, antimetabolites, such as mitomycin C and 5-fluorouracil, are considered to be the most effective ones [12,13]. However, their application is frequently associated with numerous side effects, among which are corneal toxicity, hypotony, late-onset bleb leakage, as well as infectious complications [1,3,14]. Steroids and non-steroidal anti-inflammatory drugs are also often used in order to reduce scarring [2,8]. Application of vascular endothelial growth factor (VEGF) antibodies has also been proposed for wound healing modulation and appeared to be effective in case of neovascular glaucoma. However, this strategy still did not lack undesirable side effects [15,16]. Some novel agents for wound healing modulation in glaucoma surgery are being investigated nowadays, among which are immunosuppressants with selective mechanism of action–calcineurin inhibitor cyclosporine A (CsA) and inhibitor of mechanistic target of rapamycin (mTOR) everolimus [17,18,19,20]. CsA affects T-cells causing downregulation of Interleykin-2 synthesis and, consequently, inhibition of their self-activation and proliferation, as well as indirect inhibition of macrophages and fibroblasts [21,22]. Everolimus inhibits mTOR-effects involving cell proliferation. It causes slowdown or arrest of cell cycle in G1 phase depending on its concentration. Among cells, affected by everolimus, are T-cells, smooth muscle cells, and fibroblasts [23,24,25]. 

As the above mentioned immunosuppressants are highly hydrophobic, the basis for drug delivery should also possess hydrophobic properties. A review of actual materials used in glaucoma surgery revealed that not so many commercially available glaucoma implants are lipophilic enough to cumulate either CsA or everolimus [26,27]. Among them, PLA-PEG implants turned out to be the most suitable ones due to several reasons. Firstly, this material is relatively hydrophobic allowing accumulation of hydrophobic substances. Secondly, it is highly biocompatible causing minimal reaction after implantation. Thirdly, it is biodegradable which is an advantage concerning glaucoma surgery as long-term persistence of implants can cause encapsulation and calcification [28,29]. Taking into account all of the above, we chose commercially available PLA-PEG glaucoma implants as a basis for CsA and everolimus delivery to glaucoma surgical area. Reactivity and degradation properties of these implants have been already well-studied. Their previous in vivo research concerning glaucoma surgery revealed low reactivity and low tissue response with moderate macrophages infiltration and presence of single foreign body giant cells and monocytes [30,31,32]. Their full degradation takes four to eight months and results in formation of non-toxic end products (CO_2_ and lactic acid) [31,32]. The purpose of this study was to develop a PLA-PEG-based drug delivery system with CsA or everolimus for wound healing modulation in glaucoma surgery.

## 2. Materials and Methods

### 2.1. Saturation of Glaucoma PLA-PEG Implants with CsA or Everolimus

In our study we used glaucoma PLA-PEG implants (HiBiTech, Moscow, Russia) composed of poly(rac-2-hydroxypropanoic acid)-poly(ethylene glycol) (95: 5 mass %). Implants had a shape of a folded rectangle (5.2 mm × 2.0 mm × 0.15 mm). Two commercially available versions of the implant differing in their mechanical structure were investigated in the study. One version had porous structure with porus size 30–50 μm. The second one was composed of electrospun microfibers ranging from 1.5 to 1.6 μm in diameter. 

Saturation of PLA-PEG implants with CsA was performed by means of their exposure in dilutions of commercially available CsA concentrate (50 mg/mL, Sandimmune, Novartis Pharma, Basel, Switzerland) and balanced salt solution (BSS). Investigated dilutions were the following: 1:0; 1:1; 1:2; 1:4; 1:7; 1:15 and 1:30. Exposure time was 5, 10, 15 and 30 min for each dilution. Number of implant samples used for each condition of exposure was 5. On the morrow of saturation implant samples were removed from dilutions and dried in vacuum oven at temperature 30 °C. The amount of accumulated CsA was measured by means of high-performance liquid chromatography-mass-spectrometry using chromatography system Prelude SPLC and triple-quad mass spectrometer TSQ Endura (both Thermo Fisher Scientific, Waltham, MA, USA) in gradient elution mode. Prior to the analyses implant samples were atomized and then placed in 1:1 dilution of acetonitrile and water. The mobile phase A was H_2_O with 0.1% formic acid 98–100%, mobile phase B was acetonitrile with 0.1% formic acid 98–100% (Merck Millipore, Burlington, MA, USA). Analysis was carried out in multiple reactions monitoring mode. Calibrators were prepared from dry CsA 98.5% (Sigma-Aldrich, Saint Louis, MO, USA). 

Saturation of PLA-PEG implants with everolimus was carried out by means of ultrasonic exposure of implant samples placed in everolimus suspension. Commercially available dry everolimus 95% was used (Sigma-Aldrich, Saint Louis, MO, USA). In order to prepare suspension, dry everolimus was atomized and placed in normal saline. Then it was insonated (power 630 Wt, frequency 22 kHz) for 1 min using ultrasonic disperser (Inlab, Saint Petersburg, Russia). Prior to saturation each implant sample was micro-weighed using microbalance (NETZSCH-Gerätebau GmbH, Selb, Germany). Subsequent saturation was performed with the same ultrasonic parameters in suspensions containing 1%, 2% or 3% of everolimus for 2, 4, 6 and 8 min. Then implant samples were removed from suspensions, dried in vacuum oven at temperature 30 °C and repeatedly micro-weighed. The amount of cumulated everolimus was defined as difference between weight of a sample before and after saturation.

### 2.2. In Vitro Drug Release Examination

This experimental stage involved the following implant samples (*n* = 5 for each group): porous CsA (3.5–5.0 μg spaced 0.5 μg apart), microfiber CsA (3.5 μg), porous everolimus (160–240 μg spaced 40 μg apart), microfiber everolimus (150 μg and 180 μg). All of the examined samples were placed in containers with BSS. Its volume was defined according to the volume of aqueous humor flowing through the bleb (3 mL per day; 0.75 mL per 6 h). Sink condition was definitely fulfilled both for CsA and everolimus (≥5 times more release medium). Samples were placed in incubator shaker ES-20/60 (Biosan SIA, Riga, Latvia) with rotational shaking at 50–60 rpm. Every 6 h during the first day and then every 24 h implant samples were removed from their solutions to the new ones. Concentrations of CsA in remaining solutions were detected by means of high-performance liquid chromatography-mass-spectrometry as described above. Everolimus samples were micro-weighed before and after drug release in each solution. Everolimus concentration was calculated as C = Δm/V (C—concentration, μg/mL; Δm—difference between implant weight before and after drug release, μg; V—volume of release medium, mL).

### 2.3. In Vitro Cell Experiments

The protocol was approved by the Ethics Committee of Samara State Medical University.

#### 2.3.1. Human Tenon’s Fibroblasts Primary Culture

Human Tenon’s fibroblasts (HTFs) were isolated from donors undergoing glaucoma surgery. Exclusion criteria for donors were as follows: previous ophthalmic surgery, chronic infections, history of oncology, diabetes mellitus, antimetabolites and hormones intake. All of them provided written informed consent. As surgery was performed, a piece of Tenon’s capsule tissue (2 mm × 2 mm × 1 mm) was harvested and placed in sterile shipping bottle containing sterile normal saline. After shipping to the laboratory, sterility tests were performed. Tissue samples were removed to sterile Hanks’ solution. After washing three times, samples were cut into smaller pieces and then underwent enzymatic treatment with collagenase 0.1%, which was subsequently inactivated with Versene solution 0.02% (“Biolot”, Saint Petersburg, Russia). Cells were cultured in medium 199 with 10% fetal bovine serum (“Biolot”, Saint Petersburg, Russia) and gentamycine 40 μg/ml in humidified atmosphere of 5% CO_2_ in CO_2_-incubator MCO-18AC (SanyoElectric Co, Moriguchi, Osaka, Japan). At day 21, HTFs passage was performed. At day 31, the monolayer was confluent, consisted of fibroblast-like cells with characteristic structure. This culture was used for further stages of the experiment.

#### 2.3.2. Examination of HTFs Inhibition

HTFs were cultured in 96-well plates (2 × 104 cells/well) in medium 199 with 10% fetal bovine serum. After reaching 80% confluency culture medium was replaced with new one containing immunosuppressants. CsA concentrate (Sandimmune, Novartis Pharma, Switzerland) was added to the culture medium until the demanded concentrations: 0.05 μg/mL; 0.2 μg/mL; 0.5 μg/mL; 1.0 μg/mL and 2.0 μg/mL. Control culture did not contain the drug. Dry everolimus (95%, Sigma Aldrich, St. Louis, MO, USA) was preliminary dissolved at dimethylsulphoxide (DMSO) 0.02 mL (“Biolot”, Saint Petersburg, Russia) and then added to the culture medium to reach following concentrations: 0.5 μg/mL; 1.0 μg/mL; 5.0 μg/mL; 10.0 μg/mL; 15.0 μg/mL and 20.0 μg/mL. Control culture contained 0.02 mL DMSO without everolimus. Cells were cultured in CO_2_-incubator for 7 days. 5 wells were examined for each control and study group.

Cultures were daily examined (magnification × 100, × 200) and photographed using hardware of inverted microscope “Olympics” CKX 41 («Olympus», Tokyo, Japan) and software CellSens Standart 1.7 («Olympus», Tokyo, Japan). Cells’ structural characteristics and monolayer density were analyzed. In 7 days monolayers were stained with fluorescent Live/Dead Cell-Mediated Cytotoxicity Kit in order to evaluate viability. Stained monolayers were examined using microscope “Olympus” BX41 («Olympus», Tokyo, Japan) with magnification × 100, × 200, × 400 and software CellSens Standart 1.7 («Olympus», Tokyo, Japan)and “Morphologiya 5.2” (VidioTesT, Saint Petersburg, Russia). Cytotoxicity was measured as percentage of dead cells in monolayer stained with fluorescent Live/Dead Cell-Mediated Cytotoxicity Kit.

Anti-proliferative activity of immunosuppressants was evaluated by means of proliferation index (PI), population doubling level (PDL) and doubling time (DT) calculation using following equations: PI = Nt/N_1_
PDL = (lg(Nt) − lg(N_1_))/lg2
DT = t × lg2/lg (Nt/N_1_)
where N_1_—monolayer density 24 h after medium replacement (cells/mm^2^), Nt—monolayer density at time of detection (cells/mm^2^), t—time between N_1_ and Nt measurement (hours).

### 2.4. In Vivo Study

#### 2.4.1. Animals

The study was conducted on the right eyes of 36 Soviet chinchilla rabbits weighing 3000–4000 g. Animal experiments were performed in accordance with the Code of Practice for the Housing and Care of Animals Used in Scientific Procedures and guidelines of European Animal Research Association. The protocol was approved by the Ethics Committee of Samara State Medical University. Animals were kept under appropriate conditions in special individual cages. 

#### 2.4.2. Experiment Design

All of the rabbits were divided into 3 equal groups of 12 animals each: a CsA group, everolimus group, and a control group. All of the animals underwent glaucoma filtration surgery with porous PLA-PEG glaucoma implants saturated with CsA, everolimus or not saturated with any of the drugs according to group affiliation. The follow-up period was six months during which the animals underwent slit lamp examination and tonometry. In seven days, one, and six months, four rabbits in each group were sacrificed to perform histological examination of the surgery area.

#### 2.4.3. Anesthesia 

Animals were anesthetized by intramuscular injection of fixed combination of tiletamine hydrochloride 5% and zolazepam 5% (Zoletil, Virbac Sante Animale, Carros, France) and xylazine hydrochloride 2% (Rometar, Bioveta, Ivanovice na Hané, Czech Republic) in accordance with animals’ weight.

#### 2.4.4. Implant Preparation

PLA-PEG glaucoma implants were preliminary saturated with immunosuppressants. Saturation with CsA was performed *ex tempore* in operation room just before implantation: samples were placed in 1:30 dilution of CsA concentrate (50 μg/mL, Sandimmune, Novartis Pharna, Basel, Switzerland) and BSS for 15 min. Before implantation it was dried with sterile blotting paper. Saturation of implants with everolimus was performed before sterilization by means of ultrasonic exposure (power 630 Wt, frequency 22 kHz) in everolimus suspension 2% for 6 min. 

#### 2.4.5. Surgical Technique

Limbus-based conjunctival 10 mm incision was performed. Scleral limbus-based flap was created (square-shaped, 3 mm × 3 mm). Anterior chamber was entered with 1.5 mm incision, peripheral iridectomy was performed. Implant saturated with CsA (≈4 μg), everolimus (≈240 μg) or not saturated with any drug (according to group affiliation) was placed its long side perpendicular to the limbus with one edge under the scleral flap and another one emerging above it under Tenons’ capsule and conjunctive. Then scleral flap was closed with 10/0 nylon suture (Figure 1). The conjunctival wound was closed with two 10/0 nylon sutures. After surgery animal received fixed combination of ciprofloxacin hydrochloride 0.3% and dexamethasone 0.1% (Kombinil, Sentiss Pharma Pvt. Ltd., Haryana, India) four times a day for a week.

#### 2.4.6. Pre- and Postoperative Examinations

Ophthalmic examination, including slit lamp examination and tonometry, was performed before surgery and on days 1, 3, 5, 7, 14, 28, 60, 90, and 180 following it. IOP measurements were performed with portable veterinary tonometer Tonovet (Icare Finland Oy, Vantaa, Finland). Slit lamp examination was carried out by means of SHIN NIPPON XL-1 portable slit lamp (Shin Nippon Machinery Co., Tokyo, Japan), included evaluation of overall conjunctival hyperemia (grades 0–3), bleb morphology assessment according to The Indiana Bleb Appearance Grading Scale (IBAGS). Attention was paid to possible complications–bleb leakage, corneal edema, hyphema, anterior chamber flare, cataracts, etc.

#### 2.4.7. Histologic Examination

At specific time points (7, 28, and 180) 4 randomly selected rabbits were sacrificed with overdose of anesthesia. The examined globes were enucleated together with conjunctiva and placed in 10% formaldehyde for 24 h. Then blocks containing surgery area with blebs were resected and kept in ethanol 70%. Then the resected blocks were embedded in paraffin, cut (6 μm thick) and stained with hematoxylin/eosin; hematoxylin/picro-fuchsin and hematoxylin/picrosirius red. Light microscopy examination was performed with Olympus BX-41 microscope (Olympus, Tokyo, Japan). Software “Morphologia 5.2” (VideoTesT, Saint Petersburg, Russia) was used for sections analyses. The analyses included counting and grading (from 0 to 5) collagen and cells’ density (monocytes, fibroblasts, foreign body giant cells and polymorphonucleocytes), as well as capsule thickness and neoangiogenesis.

### 2.5. Statistical Analyses

Statistical analyses were performed by means of software Statistica 12.0 (StatSoft Inc., Tulsa, OK, USA). As Shapiro-Wilk test revealed no-normal distribution numerical variables were described by median and interquartile range. Categorical variables were described by number of cases and their percentage. We used multiple linear regression analyses to describe implants saturation depending on time and concentration. Nonparametric tests were used to compare numerical variables of different groups (Mann-Whitney U-test or Kruskal-Wallis test for non-dependent variables in accordance with number of groups and Wilcoxon or Friedman tests for dependent variables in accordance with sampling). Categorical variables were compared using Chi-square test with Yates’ correction. The difference between groups was considered to be significant when *p* < 0.05. 

## 3. Results

### 3.1. In Vitro Saturation and Desorption

#### 3.1.1. Saturation of PLA-PEG Implants with CsA 

Saturation of implants with CsA in high-concentration dilutions (1:0, 1:1 and 1:3) led to their plasticization, loss of shape, and mechanical instability. The residual dilutions of CsA did not alter implant mechanical properties.

Porous PLA-PEG implants cumulated from 1.35 (1.33; 1.37) to 4.79 (4.75; 4.80) μg of CsA depending mostly on time of exposure. Drug concentration in initial dilution influenced cumulation ability to a lesser extent (Figure 2a). Microfiber samples cumulated less CsA—from 0.41 (0.40; 0.43) to 3.25 (3.22; 3.26) μg (Figure 2b). Spearmen’s rank correlations for exposure time/concentration were 0.76/0.18 and 0.74/0.40 for porous and microfiber samples, respectively. Submaximal amount of CsA was accumulated by the implants during 15 min of exposure. Further exposure did not result in significant increase of CsA amount neither for porous nor for microfiber models. 

Regression equations revealed the relationship between the amount of immunosupressants cumulated by the implants and enrichment conditions.

Porous samples of PLA-PEG glaucoma implant cumulated CsA as follows:M = −0.138 + 0.283 × t + 0.093 × c, (R^2^ = 0.95)

For microfiber samples the equation took the form:M = −0.924 + 0.194 × t + 0.186 × c, (R^2^ = 0.95)
where M—amount of cumulated CsA (μg); t—time of exposure (min; t ≤ 15 min); c—initial concentration of CsA in dilution (mg/mL; c ≤ 6.3 mg/mL).

#### 3.1.2. Saturation of PLA-PEG Implants with Everolimus 

PLA-PEG implants ability to cumulate everolimus also depended mostly on time of ultrasonic exposure. Porous samples cumulated from 131.6 (127.8; 135.3) to 248.2 (247.8; 250.6) μg of the drug (Figure 3a). Microfiber samples cumulated less everolimus: from 108.9 (105.7; 110.5) to 182.3 (180.7; 183.4) μg (Figure 3b). Spearmen’s rank correlations for exposure time/suspension concentration were 0.83/0.39 and 0.66/0.46 for porous and microfiber samples respectively. Implants cumulated submaximal amount of everolimus during first 6 min of ultrasonic exposure of porous samples and during 3 min–of microfiber ones. Further exposure did not result in significant increase of everolimus cumulation. 

Regression equation for everolimus cumulation by porous implants after ultrasonic exposure was the following (provided that time ≤ 6 min, suspension concentration ≤ 3%):M = 62.23 + 21.33 × t + 24.97 × c, (R^2^ = 0.88)

Cumulation of everolimus by microfiber samples was described by the equation (provided that time ≤ 3 min, suspension concentration ≤ 3%):M = 8.09 + 44.33 × t + 14.93 × c (R^2^ = 0.92)
where M—amount of cumulated everolimus (μg); t—time of exposure (min); c—initial suspension concentration (%).

#### 3.1.3. In Vitro CsA Release

All porous CsA implants containing ≥3.9 μg of the drug released it at concentrations exceeding 0.05 μg/mL for seven days (Figure 4). Maximal detected concentration was 1.6 µg/mL. Microfiber samples released the drug for only three days, which is not enough to provide anti-proliferative effect in glaucoma surgery. 

According to cumulative drug release profile most of CsA porous implants released 50% of cumulated CsA during the first 24 h in vitro and residual 50%-during the following 6 days. Microfiber samples released over 50% of the drug within the first 12 h and all of the residual amount within the rest of the two to five days (Figure 5).

#### 3.1.4. In Vitro Everolimus Release

All of the examined implants saturated with everolimus released the drug for 10–14 days. Similarly to experiment with CsA, porous models provided longer-term desorption than microfiber ones: 12.0 (11.0; 13.0) vs. 10.0 (10.0; 10.0) days (Figure 6). Maximal detected concentration of everolimus was 14.5 µg/mL. 

Analyses of cumulative drug release revealed that all of the samples released 45–50% of the drug within the first three days. Residual amount of the drug from porous samples was released within the following 8–10 days depending on the initial amount of everolimus cumulated by the implants. Overall drug release from microfiber samples took 10 days (Figure 7). 

Further experimental stages were carried out with samples combining the most prolonged and gradual drug release: porous 3.9 μg CsA and porous 244.5 μg everolimus implants with immunosuppressants’ desorption lasting for 7 and 13 days respectively. 

### 3.2. In Vitro Cell Culture Study

#### 3.2.1. In Vitro HTFs Inhibition

HTFs were successfully segregated from Tenon’s capsule specimens and formed confluent viable monolayer observed in a phase contrast microscope. The monolayer contained fibroblast-like cells with two to three processes. No polygonal epithelial cells were noticed in the primary culture.

HTFs cultured in the presence of CsA revealed proliferation rate 1.5–5.1 times slower in comparison with controls in dose-dependent manner. PI values measured at logarithmic phase cultures were in inverse ratio to CsA concentrations (Spearmen’s rank correlation −0.92). PDL values were significantly lower at CsA cultures compared with controls during the whole follow-up (Table 1). 

Culturing HTFs in the presence of everolimus also caused slowdown of proliferation rate by 1.7–7.4 times in comparison with controls during logarithmic phase. Unlike the case with CsA the effect was not clearly dose-dependent. PI and PDL values were significantly lower at all of the examined everolimus cultures compared with controls (Table 2). 

#### 3.2.2. In Vitro Cytotoxicity Evaluation

The percent of damaged cells in CsA and everolimus cultures did not differ significantly from control group levels. Their minimal and maximal values were 0.0–6.4%, 0.0–6.0% and 1.4–6.4% in CsA, everolimus and control groups, respectively (Figure 8). Damaged cells percentage in CsA and everolimus group was not dose-dependent. Thus, neither CsA nor everolimus revealed cytotoxic properties in examined concentrations within the range released by PLA-PEG glaucoma implants saturated with immunosuppressants as mentioned above. 

### 3.3. In Vivo Surgery Results

#### 3.3.1. Ophthalmic Examination

At day one post-op, all of the eyes were characterized by light to moderate conjunctival hyperemia which resolved after three to seven days. Light flare was observed in anterior chambers of most animals on the first day after surgery resolving gradually by day two or three. All of the blebs were diffuse and functioned well next day after surgery.

CsA and everolimus groups were characterized by slow gradual bleb shallowing during the whole follow-up period. Nevertheless, they maintained their functional activity even six months after surgery. Their height and extent, graded in accordance with IBAGS, were significantly higher in comparison with controls, ranging from one to three points in three and six months post-op. No bleb leakage was registered. Control group blebs revealed first signs of scleral-conjunctival adhesion at day seven post-op. In one month, 50% of blebs in control group were graded “zero” regarding height and extent (Figure 9, Figure 10 and Figure 11). In six months, all control blebs were flat (Figure 7). Bleb vascularity didn’t differ significantly between the groups. The Seidel test was negative for all of the animals throughout the whole follow-up.

There were two cases of implant encapsulation with severe bleb elevation and demarcation in control group in two weeks and one month post-op. As for the other complications, there were two 1-mm hyphema cases in CsA and control group (one case each). Both resolved in three days. Single conjunctival wound defects were observed in CsA and everolimus groups (one case each) at days two and three post-op. They did not reveal signs of bleb leakage and did not demand additional intervention. One case of local corneal edema coupled with surgery technique was registered in everolimus group. It spread locally in the site of unintentional corneal layers’ dissection and resolved during two weeks without special treatment. There were no signs of uveitis, cataract, endophthalmitis or other severe complications among all groups during the whole follow-up period. No significant difference in complications rate was revealed between the groups.

#### 3.3.2. Postoperative IOP Dynamics

Baseline IOP did not differ significantly between groups. Filtration surgery resulted in marked decrease of IOP levels in all of the animals. During the follow-up, IOP level in controls tended to grow gradually reaching near-baseline values. IOP increase in CsA group was much slower. By the end of follow-up, IOP values at this group were significantly lower compared both with baseline and control. The most stable hypotensive effect was observed in the everolimus group. Everolimus group IOP levels were significantly lower when compared to baseline and control. However, despite a lower median and interquartile range, no statistical differences with the CsA group were revealed (Figure 12).

#### 3.3.3. Histological Examination

Early histological examination revealed marked difference between cells’ density in the examined groups. Control group implants were densely infiltrated with monocytes, fibroblasts and foreign body giant cells. Hematoxylin/picrosirius red stain revealed denser collagen deposition inside the implants as well as thicker capsules around them in this group. Nevertheless, there were well-marked filtering spaces around control implants in early postoperative period. Less cell density was observed in drainage devices enriched with immunosuppressants tending to zero in everolimus group. Capsules around the implants were very thin in CsA group–just one to three rows of fibroblasts. Everolimus-saturated implants didn’t have capsules at all. Intensity of collagenogenesis was also much lower in CsA and everolimus groups (Figure 13). There were no signs of damage to surrounding tissues caused by immunosuppressants.

In one month, double capsules formed around control group implants, and reduction of the filtering area was observed. Capsules around implants saturated with everolimus did not adhere firmly to their material, and vast filtration spaces were marked in the surgery area. Capsules around CsA implants were thin and permeable.

By the end of follow-up surgery, the area of control group eyes was formed by connective tissue bundles infiltrated with monocytes, fibroblasts, and foreign body giant cells. Cell infiltration was much less in two other groups with marked filtration spaces around (Figure 13).

## 4. Discussion

One of the challenges in wound healing modulation is the demand for the prolonged release of anti-proliferative drugs in the surgery area. In recent decades there have been several attempts made to use calcineurin inhibitor CsA in glaucoma surgery. Most attempts, which involved single intraoperative application of the drug or postoperative topical medication with CsA emulsion, showed little or no effects in wound healing modulation [17,33]. On the contrary, prolonged administration revealed favorable effects [18,19]. It is likely that T-cells which are a target for CsA tend to appear in the wound in considerable amount only at days three to five post-op reaching their peak concentrations at days five to seven post-op. Then their number gradually declines by days 10–14 [34,35]. According to this curve, the presence of CsA in a surgical wound should be prolonged for at least seven days in order to achieve anti-inflammatory effect. The application of everolimus and its analogues has not been well studied yet. Nevertheless, according to several published studies, prolonged use of this drug is also advantageous over single application, which can also be explained by the order in which cells appear in postoperative wound [20,36,37]. T-cells reach their peak amount at days 5–7 post-op, as mentioned above, and fibroblasts reveal their growth acceleration at days 7–14 post-op [9,35]. This is why prolonged application of CsA and everolimus in wound healing modulation is reasonable in accordance with pathophysiology basis.

We chose implants composed of PLA-PEG, which possess hydrophobic properties, as a basis for delivery of CsA and everolimus to the surgery area, as these drugs are highly hydrophobic and cannot be successfully accumulated by hydrophilic materials. The other reason for such a choice was the fact that PLA-PEG implants are highly biocompatible and biodegradable with optimal degradation period ranging from four to eight months. Moreover, the chosen PLA-PEG implants have been applied in clinical practice concerning glaucoma surgery for more than ten years and have been studied well. Their application caused little complications and enhanced the effectiveness of glaucoma surgery [30,31,32]. 

Within our research, we developed a method and defined conditions most suitable for saturation of PLA-PEG glaucoma implants with CsA and everolimus. 

PLA-PEG implants accumulated CsA by means of simple exposure in CsA dilutions without any additional procedures, as a result of which porous samples could hold up to 4.95 (4.93; 5.00) μg of CsA. The period of in vitro drug release for porous implants containing 3.9–5.0 μg of CsA was seven days. These results correspond with published data stating that PLA-based materials are good at long-term release of many drugs, including CsA [18,38]. One of the advantages of our method is the possibility to perform it ex tempore without any special equipment. Moreover, developed math models, forecasting the amount of CsA cumulated by PLA-PEG implant, depended on time of exposure and dilution concentration, makes it possible for a surgeon to regulate the amount of CsA cumulated by the implant. 

Though it is possible to achieve longer CsA release by means of its mixture with initial polymer solution before formation, the achieved seven-day drug release is not insufficient. On the contrary, it is fully approved with regard to pathophysiology of wound healing as mentioned above. It is claimed that at 10–14 days post injury regulation, the mechanisms of wound repair switch from activation to inhibition of T-cells role in wound repair [34]. Consequently, their additional inhibition is no longer needed at this period of time.

PLA-PEG glaucoma implants were also characterized by good everolimus accumulation and prolonged drug release. This corresponds with published data and wide clinical use of PLA-based materials for prolonged everolimus release (for example, in drug eluting stents production). Unlike the case with CsA, simple exposure in everolimus suspension was not enough for saturation. A needed condition for implants’ enrichment with everolimus was ultrasonic exposure, as a result of which porous samples could hold up to 244.5 (241.8; 245.3) μg of everolimus. 

It is likely that due to lower solubility in water and higher initial drug content, everolimus release was almost twice as long as CsA one (i.e., lasting up to 13 days). Such prolonged drug release time is suitable as it overlaps with the period of the most intense T-cell activation and fibroblasts’ acceleration (which are the main effector cells) and does not embrace the period when regulatory immune mechanisms start to limit wound healing. 

Little data are published in regards to everolimus application in glaucoma surgery [20], but the study investigating prolonged drug delivery of sirolimus (also mTOR inhibitor similar to everolimus) revealed promising results [36]. 

In our experiment, porous implants revealed a better ability to cumulate both CsA and everolimus as well as their longer desorption in comparison with microfiber models which could be coupled with the difference in their structure and density. 

In vitro cell experiment was carried out in order to define if concentrations of CsA and everolimus released by the implants in vitro cause anti-proliferative effect and if they reveal cytotoxicity. According to the data, published earlier, CsA affects the viability of the most cells of the eye (corneal epithelium, corneal endothelium, retinal pigment epithelium) at concentrations exceeding 5 μg/mL, causing serious damage at concentratios higher than 50.0 μg/mL [38,39,40,41]. In our experiment maximal fixed CsA concentration was 1.6 μg/mL, which is much lower than mentioned above levels. But little is known about CsA effect on HTFs. Few data have been published about everolimus influence on HTFs and other cells of the eye. Data scattering in regard of anti-proliferative and toxic concentrations of everolimus is quite big, ranging from nanograms to micrograms [20,42,43,44]. That was the main rationale for the reported experimental stage.

In our experiment, neither CsA nor everolimus caused cytotoxic effects in concentrations released by the implants. Both examined immunosuppressants caused slowdown of HTFs proliferation by a factor of 1.5–5.1 and 1.7–7.4 for CsA and everolimus, respectively. 

It is worth noting that CsA caused anti-proliferative effect on HTFs in cell culture lacking cellular and cytokine environment. This implies that CsA somehow causes direct anti-proliferative effect on HTFs bypassing T-cells which are considered to be the main therapeutic target of this drug. This corresponds to the data, published earlier by A. Leonardi [45] and M. Viveiros [46], who also described direct influence of CsA on fibroblast proliferation. 

In our in vivo experiment, the application of PLA-PEG implants saturated with either CsA or everolimus improved glaucoma filtration surgery outcomes when compared to surgery with the same implants, but without immunosuppressants. This consisted in the improvement of bleb function and morphology, as well as a better hypotensive effect. Histological examination corresponded with clinical results. It is of no small importance that the application of implants saturated with CsA or everolimus did not cause acceleration of complications rate in comparison with control group. Earlier studies, dedicated to single intraoperative CsA application or postoperative topical treatment with CsA drops, did not demonstrate any improvement of glaucoma surgery outcomes. Nevertheless, it has already been reported that prolonged CsA application enhances the effectiveness of glaucoma surgery involving tube drainage device [18]. Inhibitors of mTOR, though not well-studied in glaucoma surgery yet, also revealed better results in the case of long-term application compared to single administration [36]. Our results also suggest that prolonged CsA or everolimus delivery represents a reasonable approach in wound healing modulation in glaucoma filtration surgery. 

## 5. Conclusions

Thus, there was developed a PLA-PEG-based drug delivery system implying prolonged release of cyclosporine A or everolimus for 7 and 13 days, respectively. Either CsA or everolimus revealed proven anti-proliferaive but not cytotoxic effects on HTFs in vitro at concentrations released by the developed system. Application of PLA-PEG glaucoma implants saturated with CsA or everolimus improved glaucoma filtration surgery outcomes without acceleration of complications rate due to wound healing modulation.

## Figures and Tables

**Figure 1 polymers-14-03419-f001:**
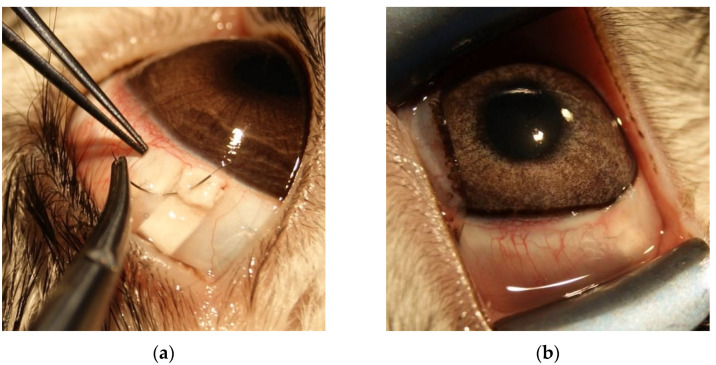
Surgical technique. (**a**) PLA-PEG implant with edge under the scleral flap, which is being fixed; (**b**) Post-op bleb. Proximal edge of the implant is visible through conjunctiva.

**Figure 2 polymers-14-03419-f002:**
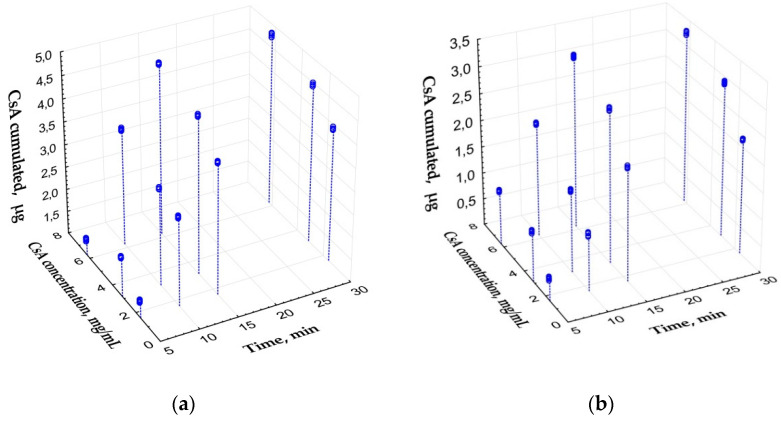
Scatterplot of the amount of cumulated CsA depending on enrichment conditions (time and CsA concentration in initial dilutions). (**a**) Porous samples. (**b**) Microfiber samples.

**Figure 3 polymers-14-03419-f003:**
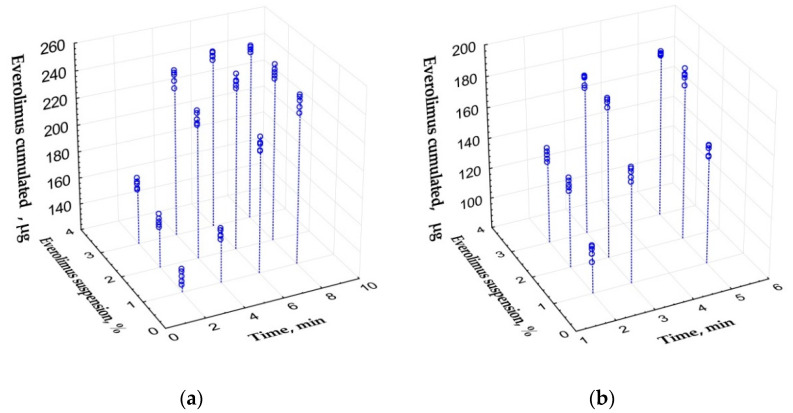
Scatterplot of the amount of cumulated everolimus depending on enrichment conditions (time and everolimus suspension concentration in initial dilutions). (**a**) Porous samples. (**b**) Microfiber samples.

**Figure 4 polymers-14-03419-f004:**
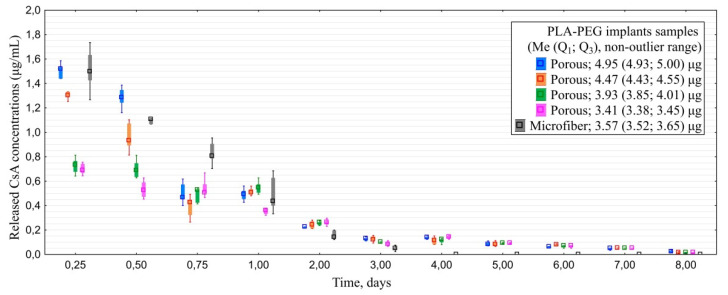
Dynamics of CsA release from PLA-PEG implants samples depending on their structure and initial CsA amount.

**Figure 5 polymers-14-03419-f005:**
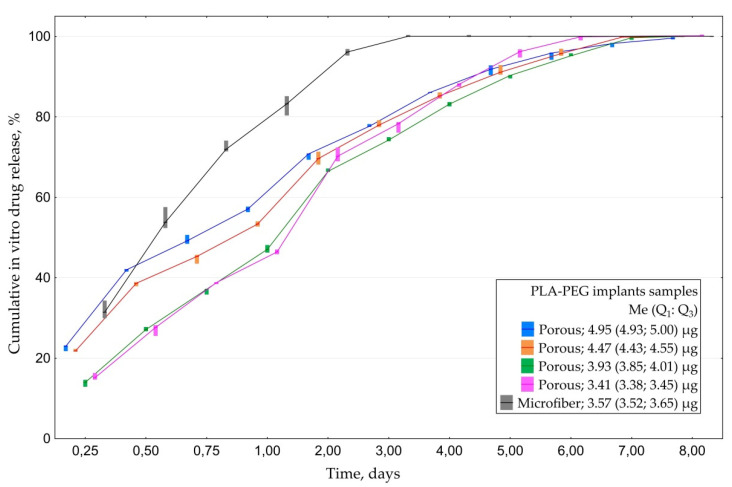
Cumulative drug release of CsA from PLA-PEG implants samples depending on their structure and initial CsA amount.

**Figure 6 polymers-14-03419-f006:**
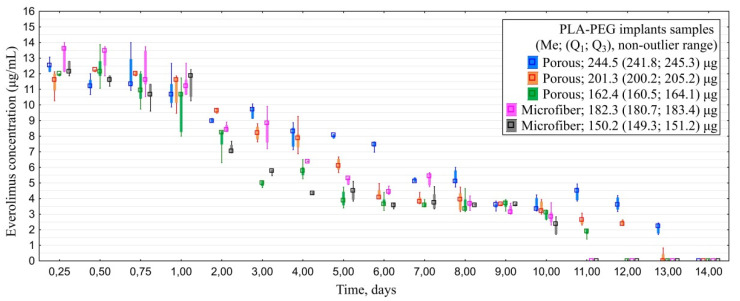
Dynamics of everolimus release from PLA-PEG implants samples depending on their structure and initial everolimus amount.

**Figure 7 polymers-14-03419-f007:**
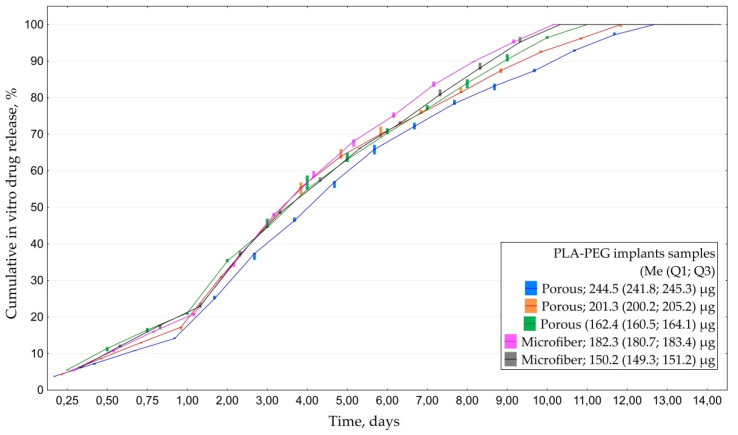
Cumulative drug release of everolimus from PLA-PEG implants samples depending on their structure and initial drug amount.

**Figure 8 polymers-14-03419-f008:**
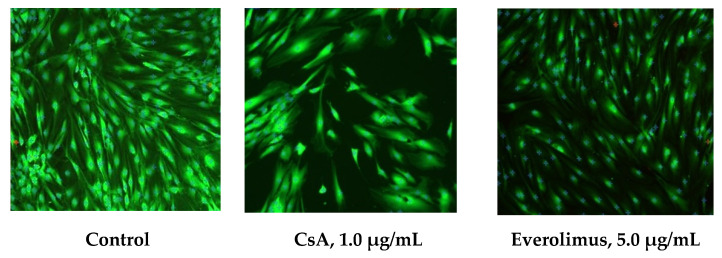
Fluorescence microscopy of HTFs cultures. Day seven. Stain: Live/Dead Cell-Mediated Cytotoxicity Kit. Magnification × 100. Calculation of damaged and viable cells using software CellSens Standart 1.7.

**Figure 9 polymers-14-03419-f009:**
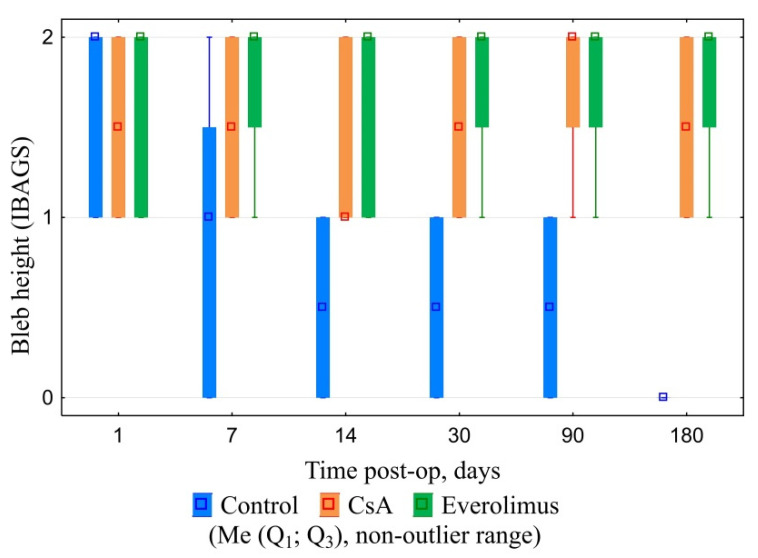
Blebs’ height dynamics.

**Figure 10 polymers-14-03419-f010:**
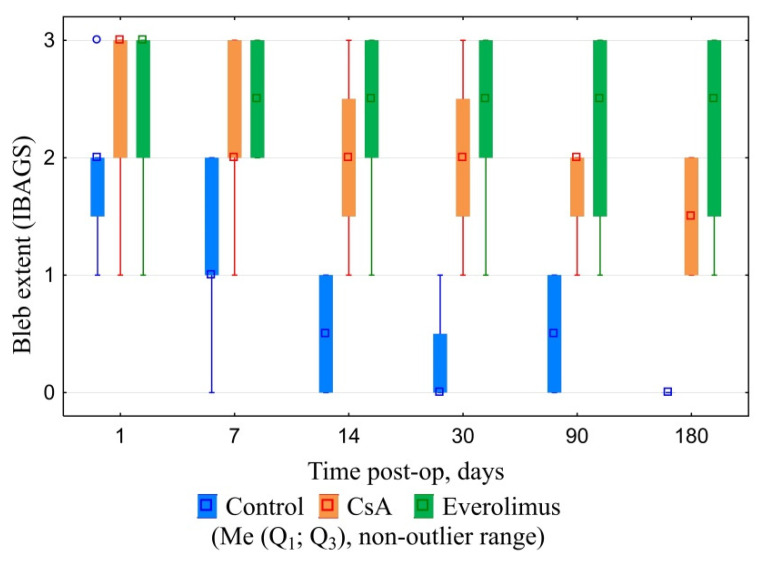
Blebs’ extent dynamics.

**Figure 11 polymers-14-03419-f011:**
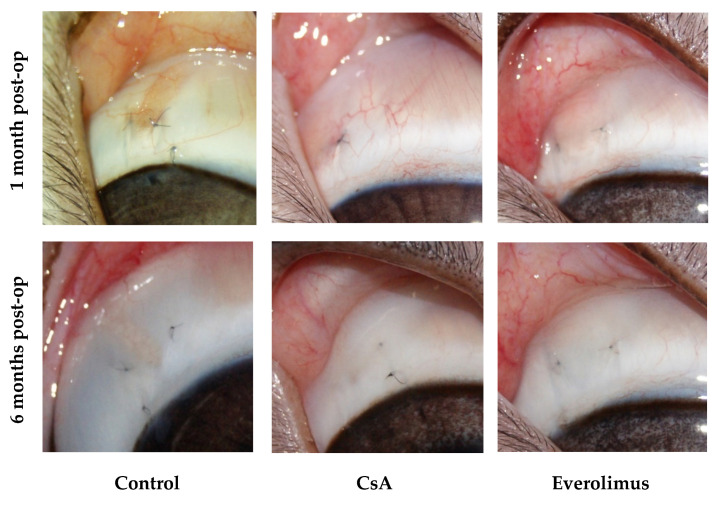
Anterior segment of rabbit eyes one and six months post-op. Surgery area. Blebs.

**Figure 12 polymers-14-03419-f012:**
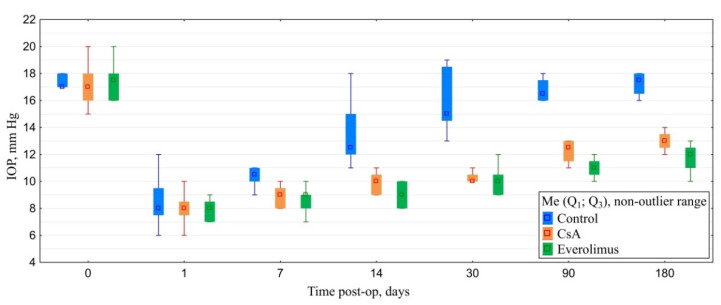
IOP dynamics.

**Figure 13 polymers-14-03419-f013:**
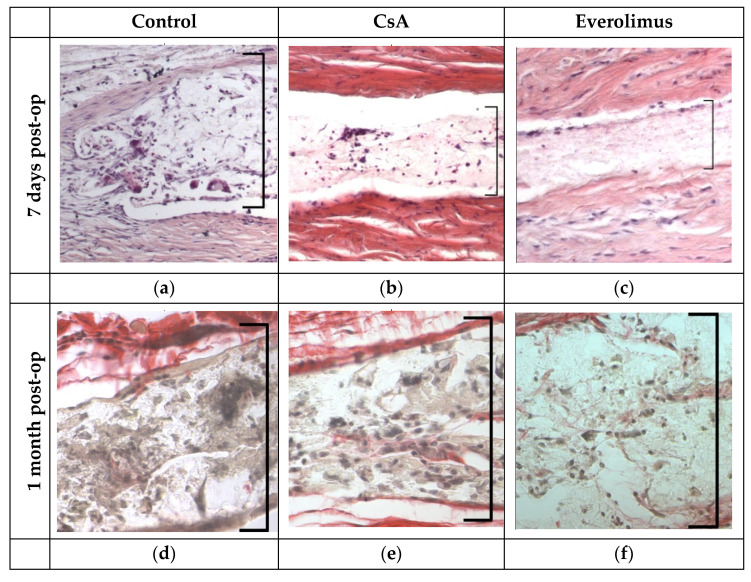
Histological sections of surgery area with implants. Square bracket indicates the implant area. (**a**–**c**). Stain: hematoxylin/eosin. Magnification × 100. (**d**–**f**). Stain: hematoxylin/picrosirius red. Magnification ×400. (**g**–**i**). Stain: hematoxylin/eosin. Magnification ×400.

**Table 1 polymers-14-03419-t001:** HTFs inhibition in control (0.00 μg/mL) and CsA (0.05–2.00 μg/mL) groups, Me (Q_1_; Q_3_).

CsA Conc, μg/mL	PI, rel. un.	DT, h	PDL, PI, rel. un.
0.00	2.35 (1.92; 2.36)	39.0 (38.7; 50.9)	1.55 (1.53; 1.56)
0.05	1.70 (1.63; 1.75) *	62.9 (59.5; 67.8) *	0.96 (0.91; 0.97) *
0.20	1.52 (1.41; 1.56) *	79.2 (75.3; 96.6) *	0.75 (0.73; 0.76) *
0.50	1.46 (1.41; 1.48) *	88.1 (84.9; 97.8) *	0.62 (0.54; 0.62) *
1.00	1.38 (1.37: 1.41) *	102.4 (97.4; 107.0) *	-
2.00	1.22 (1.17; 1.29) *	182.5 (155.0; 208.7) *	0.70 (0.55; 0.75) *

* *p* < 0.05 when compared to control (0,00 μg/mL). PI—Proliferation index, PDL—Population doubling level, DT—doubling time.

**Table 2 polymers-14-03419-t002:** Proliferation indices of HTFs cultures of control (0.0 μg/mL) and everolimus (0.5–20.0 μg/mL) groups, Me (Q_1_; Q_3_).

Everolimus Conc, μg/mL	PI, rel. un.	DT, h	PDL, rel. un.
0.0	1.96 (1.94; 1.99)	49.6 (48.2; 50.0)	1.32 (1.31; 1.49)
0.5	1.43 (1.21; 1.46) *	93.1 (87.7; 175.0) *	0.63 (0.40; 0.67) *
1.0	1.23 (1.22; 1.23) *	162.1 (161.3; 168.8) *	0.41 (0.41; 0.42) *
5.0	1.17 (1.07; 1.22) *	188.3 (145.5; 342.3) *	0.51 (0.51; 0.60) *
10.0	1.13 (1.12; 1.14) *	279.7 (262.3; 289.1) *	0.41 (0.32; 0.54) *
15.0	1.29 (1.28; 1.31) *	129.3 (124.7; 133.8) *	0.73 (0.65; 0.79) *
20.0	1.54 (1.37; 1.59) *	77.1 (72.2; 104.6) *	0.85 (0.68; 0.87) *

* *p* < 0.05 when compared to control (0.0 μg/mL). PI—Proliferation index, PDL—Population doubling level, DT—doubling time.

## Data Availability

Data supporting reported results are available at request.

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
