# Peer review of "PLA-PEG Implant as a Drug Delivery System in Glaucoma Surgery: Experimental Study"

_polymers, 2022, doi:10.3390/polym14163419_

Round 1
Reviewer 1 Report
The manuscript demonstrated a PLA-PEG-based drug delivery system with cyclosporine A or everolimus. The experiments clearly suggested that the PLA-PEG system shows long-term drug release and the system is available for wound healing in glaucoma surgery. The manuscript seems to provide enough evidence that establishes the author’s statements.
Major points
The introduction does not show any information about PLA-PEG. Advantages of using of PLA-PEG and recent studies for PLA-PEG, such as a second paragraph of the discussion section, should be shown in the introduction.
I’m not certain why the authors use two versions of structures of PLA-PEG. Can the faster-release of drugs for microfiber be connected with the mechanical structure?
Minor points
3.1.3. In vitro everolimus release → 3.1.4.
Line 312-313 “All the examined implants…”: The sentence should be written at the next section; “3.1.4. In vitro everolimus release”. The authors should not compare the results of the CsA release with the everolimus release before the explanation of figure3.
Reviewer 2 Report
The authors develop a PLA-PEG-based delivery system with cyclosporine A or everolimus to promote wound healing after glaucoma surgery. The paper is well organized and shows a promising strategy of wound healing modulation in glaucoma filtration surgery. However, there are some major concerns that need to be addressed before acceptance.
1. The paper only mentions the advantages of PLA but does not explain why PLA-PEG was used as implant materials.
2. Please give the full terms of the abbreviations, e.g., BSS.
3. Please give detailed information about the drug loading. The authors only give the cumulated drug amounts after different exposure times (n=2), which is insufficient.
4. Please add the experiments to evaluate the in vitro and in vivo degradation properties of PLA-PEG implants.
5. A cumulative drug release profile could also be calculated and drew to better explain drug release characteristics from implants.
6. The drugs were loaded into the PLA-PEG implants in this study. But the in vitro cell study was conducted by directly incubating the cells with free drugs. It should be done by incubating the drug-loaded PLA-PEG implants with the cells.
7. The authors should indicate the implant area in the HE images if there is.
8. The authors claimed that the PLA-PEG only released the drug in a sustainable manner for about 3~13 days. But the in vivo study lasted about six months. So whether the PLA-PEG blank implants could affect the therapeutic outcomes?
9. Please make sure the reference format meets the Journal requirement.
